# ARE YOU GOING TO FINISH THAT? A PRACTICAL STUDY OF THE TOKENIZATION BOUNDARY PROBLEM

## ABSTRACT

Language models (LMs) are trained over sequences of tokens, whereas users interface with LMs via text. When a user (unknowingly) ends their prompt in the *middle* of the expected next-token, the predicted next-token distribution becomes distorted. While this phenomenon has been extensively documented in prior work using arbitrary character prefixes, less attention has been paid to how often it occurs in realistic prompts that adhere to word boundaries, or whether the distortion persists in these cases. In this work, we identify three domains where token boundaries commonly do not line up with semantic or syntactic ones: languages that do not use whitespace, highly compounding languages, and code. For instance, we find that in Chinese text, up to 25% of word boundaries do not line up with any token boundary, meaning that even prompts ending with complete words are susceptible to probability distortion. We then systematically construct semantically natural prompts that end with a partial token and measure the effect on predictions. We find that these constructions comprise a serious failure mode: frontier LMs consistently place two orders of magnitude less probability on the correct continuation compared to when the prompt is "backed-off" to be token-aligned, despite being given strictly more context. Moreover, this phenomenon exhibits inverse scaling, with probability distortion increasing for larger models. Finally, we evaluate `ByteSampler`, a recently proposed sampling-time fix for the tokenization boundary problem, and find that it effectively and efficiently overcomes the problem, exceeding the performance of heuristic token backoff. Overall, we demonstrate the scale and severity of probability distortion caused by tokenization in realistic use cases, and recommend that model inference providers adopt an inference-time fix by default at every prompt boundary.

## 1 INTRODUCTION

Users of language models (LMs) generally interface with LMs via text, providing a string as a prompt and expecting a string continuation. However, modern LMs actually operate over **sequences of tokens**, which are produced by a tokenizer that segments these **strings of characters** into multi-character units for more efficient processing. This basic mismatch means that users may (unknowingly) provide a prompt that ends with the prefix of a valid token, which then causes the model to assign unexpectedly *low* probability to the completion of that token.

This seemingly simple issue is what we call the *tokenization boundary problem* (TBP).[1] It underlies (for example) the standard advice to users to not end prompts with a trailing whitespace, as the whitespace would be a partial token relative to the many tokens in the LM vocabulary that start with a whitespace, and has empirically produced severely degraded results. While the TBP has been extensively documented in prior work (Lundberg, 2023; Ribeiro, 2023; Phan et al., 2024) and many methods have been proposed to address it (Athiwaratkun et al., 2024; Vieira et al., 2025; Phan et al., 2025; Turaga, 2025; Hayase et al., 2025), these works generally illustrate the TBP using arbitrary character prefixes of the text, which may not reflect realistic user prompts. As a result, the extent to which TBP affects natural use cases remains unknown.

---

[1] In other works, this issue has been called the "prompt boundary problem" or "tokenization bias," but we choose to rename it to the *tokenization* boundary problem both to highlight the role of tokenization and also to accurately reflect that the boundary problem may also occur at the end of the continuation, not just at the end of the prompt. (See next footnote)

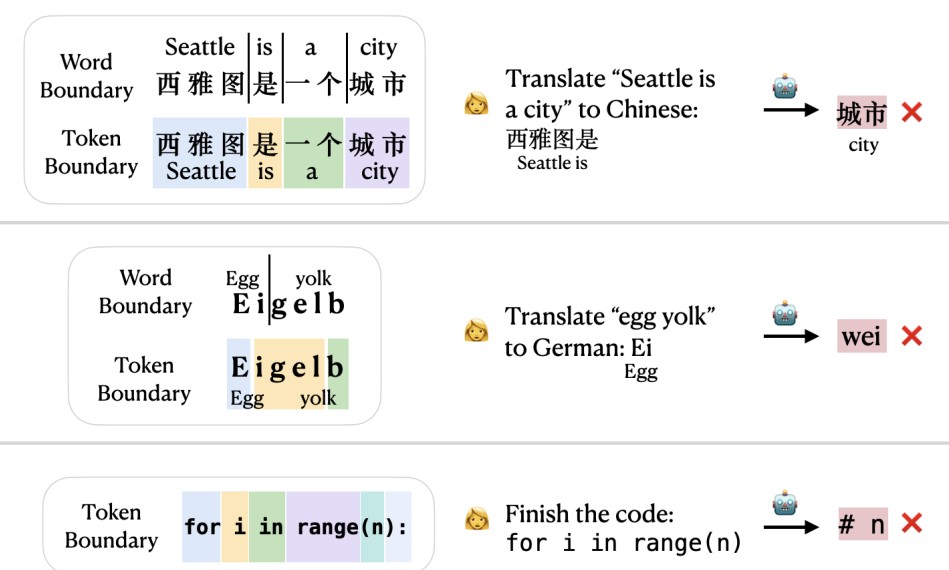

Figure 1: Examples of mismatches between linguistic units and tokens. **Upper:** example of Chinese. The model cannot correctly generate token 一个(a) if the prompt ends naturally after word 是(is). **Middle:** example of German. The model cannot generate correct translation of word "egg yolk". **Lower:** example of code completion. Model continuations are generated by QWEN3-4B.

This work is the first to quantify the severity of the TBP in natural and realistic prompts, including both its prevalence and the level of distortion it wreaks on next-token probability distributions. We identify three domains where whitespace does not reliably separate semantic or syntactic units (see illustrations in Fig. 1), so that even prompts with complete word endings would be susceptible to the boundary problem:

- **Logographic writing systems.** Unlike languages such as English or French, where words are typically separated by whitespace, words in languages such as Chinese and Japanese are not delimited. Tokenizer training algorithms can therefore merge characters across word boundaries, and token and word boundaries commonly do not align. For instance, "是/一个" (*is / a*) is tokenized as the single token 是一个 in the QWEN3 tokenizer, meaning that a prompt that ends with the complete word "是" (is) would be affected by the TBP.

- **Languages with highly productive compounding**. In German, words are often built by combining two or more full words. Thus, similarly to Chinese, a tokenizer can segment compound words in ways that do not respect morphological boundaries. For example, *Eigelb* (*egg yolk*) comes from *Ei* (*egg*) and *gelb* (*yellow*). LLAMA3 tokenizes "␣*Eigelb*" as ⟨␣E, igel, b⟩, so a prompt ending with "␣*Ei*" would similarly be affected by the TBP.

- **Code completion.** Code often contains long identifiers or consecutive punctuation that are merged into single tokens. For example, in the Python main function definition "*def main():*", `():` is a single token. As a result, the TBP affects prompts that ends with the punctuation "*()*".

We develop a methodology that systematically constructs prompts that align with natural word boundaries but end with a partial token, where the completion of that token is the unambiguous continuation, in order to create controlled experimental conditions that expose the TBP. We find across a wide range of models that the TBP negatively impacts both accuracy and model confidence in predictions. For instance, in Chinese, the probability assigned to the correct next-token drops by four *orders of magnitude* in the token-misaligned case compared to the aligned case. The accuracy of the continuation drops by 30% – 70% across all models and domains. Surprisingly, this behavior consistently exhibits inverse scaling across model families, where larger models suffer *more* from the TBP, likely because larger models are more fit to their tokenizer.

Fortunately, there is a solution to the TBP. Hayase et al. (2025) recently introduced `ByteSampler`, which constructs a tree of all possible valid token sequences that "cover" the provided prompt, then selects the path from root to leaf with the greatest cumulative log-probability, We apply `ByteSampler` to our dataset, finding that it solves the problem completely, exceeding the performance of heuristic token backoff by over 15%. It is also efficient, using $\sim 1$ additional forward pass on average to move past the prompt boundary, after which sampling can resume in the normal way. We therefore recommend that all deployments of tokenization-based LMs use `ByteSampler` to correct for TBP distortion at the end of the user prompt.

Overall, our work demonstrates how a fundamental problem caused by tokenization surfaces in realistic use cases, and provides new evaluations of a complete solution.

## 2 BACKGROUND ON TOKENIZATION BOUNDARY PROBLEM

The task of language modeling is to define a probability

$$P(\text{continuation} \mid \text{prompt})$$

for given prompt, over all possible continuation strings. In practice, tokenizer-based LMs model the above by encoding the text as a sequence of tokens and autoregressively calculating the probability of each subsequent token,

$$P(c_1, ..., c_n \mid p_1, ..., p_m) = \prod_{i=1}^{n} P(c_i \mid p_1, ..., p_m, c_1, ..., c_{i-1}) \qquad (1)$$

where $\langle p_1, ..., p_m \rangle = \text{encode}(\text{prompt})$ and $\text{decode}(\langle c_1, ..., c_n \rangle) = \text{continuation}$. Because LMs are trained only on valid token sequences, by which we mean token sequences in the output space of the encode function, the prediction of Equation 1 is only properly-conditioned when the concatenation of the prompt and continuation, tokenized individually, $\langle p_1, ..., p_m, c_1, ..., c_n \rangle$, is also a valid token sequence, or in other words,

$$\text{encode}(\text{prompt}) + \text{encode}(\text{continuation}) = \text{encode}(\text{prompt} + \text{continuation}) \qquad (2)$$

where we use $+$ to denote both string and list concatenation. Note that Equation 2 is not true, in general, because the prompt ending forces a token boundary that may or may not be present when the prompt and continuation are tokenized together (right side). When Equation 2 is not satisfied, the *tokenization boundary problem* (TBP) may arise. In this case, there is a token $T$ from $\text{encode}(\text{prompt} + \text{continuation})$ that bridges the prompt-continuation boundary, but is not present in that position in $\text{encode}(\text{prompt}) + \text{encode}(\text{continuation})$. Formally, a prompt suffers from the TBP when the *probability* of the sequence $\text{encode}(\text{prompt}) + \text{encode}(\text{continuation})$ (which contains a token boundary forced by the prompt ending) differs from that of $\text{encode}(\text{prompt} + \text{continuation})$ (the canonical tokenization of the entire string).

A practical (if contrived) English example illustrates the TBP with LLAMA3-3.2-1B. Suppose we have prompt = "␣*natural*␣*language*␣*processin*", and continuation = "*g*". In this case, Equation 2 is not satisfied, as

$$\text{encode}(\text{prompt} + \text{continuation}) = \langle \texttt{␣natural}, \texttt{␣language}, \texttt{␣processing} \rangle$$
$$\text{encode}(\text{prompt}) + \text{encode}(\text{continuation}) = \langle \texttt{␣natural}, \texttt{␣language}, \texttt{␣process}, \texttt{in}, \texttt{g} \rangle$$

Here, the token $T = \texttt{␣processing}$ bridges the prompt-continuation boundary. To verify the effect of the TBP, we observe that, under the model's distribution:

$$P(\text{encode}(\text{prompt} + \text{continuation})) = 4.77 \times 10^{-8}$$
$$P(\text{encode}(\text{prompt}) + \text{encode}(\text{continuation})) = 1.58 \times 10^{-15}$$

which differ substantially despite corresponding to the same string.[2] In particular, the probability of token $\texttt{g}$ when conditioned on "␣*natural*␣*language*␣*processin*" is only 0.002, even though it is overwhelmingly the most reasonable continuation.

---

[2]In order to set up notation for the experiments, we have focused on the scenario where the TBP occurs at the end of the prompt. However, the TBP can also occur at the end of the continuation. For instance, the string "␣*processin*" given "␣*natural*␣*language*" has lower likelihood ($1.35 \times 10^{-5}$) under LLAMA3.2 than "␣*processing*" given the same context (0.788), despite "␣*processin*" being a substring of "␣*processing*". This is because the likelihood of the token sequence $\text{encode}(\text{"␣}processin\text{"})$ actually *excludes* the probability of the string "␣*processing*", for which the single token $\texttt{␣processing}$ would be used.

Fortunately, the TBP is generally avoidable in English and other whitespace-delimited languages. This is because the *pretokenization step* in tokenizer training algorithms split text on whitespace, preventing merges across it; in particular, since splits occur to the left of whitespace, learned tokens can only have *leading* (not trailing) whitespace. By ending prompts with complete words (with no trailing whitespace), user prompts can line up with the token boundary.

Many methods have been proposed to address the TBP. For instance, the heuristic technique of token healing (Ribeiro, 2023; Dagan et al., 2024; Athiwaratkun et al., 2024) "backs off" the prompt by removing one or more tokens from the prompt ending, then sampling a continuation that is constrained to match the removed text. In contrast, exact methods preserve the probability distribution over text of the original LM (Vieira et al., 2025; Phan et al., 2025; Turaga, 2025; Hayase et al., 2025). In both cases, these works generally evaluate the TBP using arbitrary character prefixes of the text, which may not reflect realistic user prompts. In this work, we instead study how commonly the TBP arises in practice and whether the probability distortion persists when prompt endings align with natural word or syntactic boundaries.

## 3 MISALIGNMENT BETWEEN TOKENS AND SEMANTIC/SYNTACTIC UNITS

First, we wish to quantify how likely probability distortion at the prompt boundary might occur in real use cases. We define the **misalignment rate** as the proportion of word boundaries (or syntactic boundaries, in the case of code) that do not line up with a token boundary. We consider three domains where whitespace does not reliably separate semantic or syntactic units, such that even prompts with complete word endings may end with partial tokens.

**Chinese**   We sample 1,000 entries from Chinese Wikipedia and obtain word boundaries with the off-the-shelf segmentor `Jieba`, a widely-used Chinese word segmentor based on a prefix dictionary.

**German**   We sample 1,000 entries from German Wikipedia and obtain word boundaries with `CharSplit`, an $n$-gram-based compound splitter built specifically for German.

**Code**   Since each punctuation character in code usually delimits some syntactic boundary (e.g., the closing a parenthesis), we consider the misalignment rate to be the proportion of right-boundaries of punctuation characters that lie in the middle of a token. We sample 200 snippets for each of six programming languages from the CodeXGLUE dataset (Husain et al., 2019).

| | Chinese | | German | Code | | | | | |
|---|---|---|---|---|---|---|---|---|---|
| | **Vocab** | **Wiki** | **Wiki** | **Python** | **Java** | **JS** | **Go** | **PHP** | **Ruby** |
| QWEN 3 | 23768 | 19.96 | 6.51 | 68.08 | 65.31 | 61.35 | 55.75 | 52.52 | 62.41 |
| DEEPSEEK-V3 | 18052 | 23.43 | 4.38 | 63.55 | 60.66 | 57.90 | 52.07 | 50.09 | 55.61 |
| HUNYUAN | 45431 | 24.92 | 37.20 | 66.28 | 63.53 | 59.78 | 53.23 | 51.32 | 59.07 |
| LLAMA 3 | 4301 | 24.50 | 37.84 | 68.08 | 65.31 | 61.35 | 55.75 | 52.52 | 62.41 |
| OLMO 2 | 313 | 14.88 | 8.09 | 68.08 | 65.31 | 61.35 | 55.75 | 52.52 | 62.41 |
| MISTRAL NEMO | 2963 | 18.10 | 5.46 | 65.67 | 62.05 | 59.33 | 52.86 | 50.99 | 57.87 |
| GEMMA | 18910 | 14.12 | 3.40 | 25.11 | 24.90 | 23.81 | 16.21 | 32.77 | 16.98 |

Table 1: **There is a high rate of misalignment between token units and semantic/syntactic units** when using off-the-shelf tokenizers to encode Chinese, German, and code data, where zero misalignment means that semantic/syntactic boundaries never occur in the middle of a token. Misalignment creates the potential for the tokenization boundary problem to affect even natural prompts that respect word and syntactic boundaries.

Shown in Table 1, tokenizers produce a high misalignment rate across the board. In Chinese, 14% and 25% of word boundaries do not lie on a token boundary. Prompts ending along these misaligned word boundaries would not be properly-conditioned, resulting in a distorted next-token prediction. We observe that this misalignment rate is not necessarily tied to the number of Chinese tokens in the vocabulary: LLAMA 3 has only 4,301 Chinese tokens but shows the second highest misalignment rate of 24.5%, while HUNYUAN, with a much greater set of 45,131 Chinese tokens, shows a

| Prompt | Expected cont | Predicted cont |
|---|---|---|
| It is
它是 | a  island
一个群岛 | island
群岛 |
| It
它 | is a    island
是一个群岛 | is a    island
是一个群岛 |
| It commonly appears
它 经常 出现 | in
在 | and
于 |
| It commonly
它 经常 | appears in
出现在 | appears in
出现在 |
| The largest part of the route is with stone
Der größte Tiel des Wegverlaufs ist mit Stein | steps
stufen | plates
platten |
| The largest part of the route is with sto...
Der größte Tiel des Wegverlaufs ist mit Ste | ne steps
instufen | ne steps
instufen |
| The easiest way to an introduction in the travel
Der einfachste Weg um einen Einstieg in die Reise | literature
literatur | reporter
berichter |
| The easiest way to an introduction in the trav...
Der einfachste Weg um einen Einstieg in die Re | el literature
iseliteratur | el report
isebericht |
| `def _` | `_init__` | `__(self)` |
| `def` | `__init__` | `__init__` |
| `def reverse_words(` | `s)` | `____)` |
| `def reverse_words` | `(s)` | `(s)` |

Table 2: **Examples from our dataset.** In each row, the *word-aligned prompt* is shown on top, where we use red to highlight the text corresponding to the token $T$ that is truncated prematurely by the prompt ending. The *token-aligned prompt* is shown on the bottom, which backs off the prompt to the token boundary preceding $T$. The predicted continuation is shown for QWEN3-32B.

similar misalignment rate of 24.9%. The misalignment in German is lower, as whitespace is used in conjunction to word compounding. For code, punctuation characters commonly lie in the middle of longer tokens, at a rate of $\geq 50\%$ across all tokenizers and programming languages, except for GEMMA's tokenizer at 16.98%. This high misalignment rate is unsurprising as punctuation characters commonly occur contiguously in code, and most commonly used pretokenizers allow punctuation to be merged together freely. Note that we focus on syntactic boundaries in code to be analogous with the natural language setting, but in the popular use case of code autocompletion, users often pause their coding anywhere, including in the middle of function names or identifiers that would not represent a syntactic boundary.[3] Thus our definition of misalignment is a lower bound on how often the TBP would affect realistic use cases.

## 4 EXPERIMENTS

Next, we construct natural prompts in Chinese, German, and code that expose the TBP, and measure the resulting effect on model predictions. See Table 2 for example prompts.

### 4.1 PROMPT CONSTRUCTION

Our goal is to create (prompt, continuation) pairs that satisfy three criteria: (1) the prompt is semantically and syntactically natural, (2) the continuation is unambiguous, ensuring accurate evaluation, and (3) tokenizing the prompt + continuation together would produce a token spanning the prompt-continuation boundary (i.e., Equation 2 is violated).

To satisfy conditions (1) and (3), we segment sentences into both word and token units, and identify word boundaries that do not align with token boundaries. We then split the sentences at these positions to form prompts and continuations. To ensure the continuation is unambiguous (condition 2), we use translation tasks for Chinese and German, and provide task instructions for code. Each

---

[3]In fact, Cursor describes this exact problem and calls for solutions in a blog post from January 2025.

model is evaluated on test set constructed with its own tokenizer, so dataset sizes vary slightly due to different word-token misalignment rates. We describe each domain in more detail below.

**Chinese**  We frame the task as English-to-Chinese translation using parallel corpus FLORES (NLLB Team et al., 2022). Chinese sentences are segmented with `Jieba` for words and the model's tokenizer for tokens. Prompts present the English source and request a Chinese translation, with the target translation partially provided. Using all 997 FLORES entries, we extract cases where word and token boundaries misalign, treating each mismatch as a separate test case. Thus, a single sentence may create multiple test entries. Test set varies by tokenizer, as shown in Table 3.

**German**  We follow the same approach as for Chinese: we frame the task as translation from English to German, use the parallel corpus FLORES, and use the segmentor `CharSplit`.

**Code**  For code, we draw examples from MBPP (Austin et al., 2021). We end the prompt inside multi-character punctuation tokens, which usually represents a syntactic boundary. For example, in a Python function declaration "*def main():*" where `():` is a single token, we truncate the test case to be "*def main()*". The prompt consists of the docstring, which specifies the desired functionality, followed by a prefix from the solution code. We randomly sample 300 entries from the MBPP dataset and extract all cases where a punctuation mark falls inside a single token, following the same procedure as for Chinese and German.

## 4.2 METRICS

In §4.1, we created $(\text{prompt}, \text{continuation})$ pairs. We denote $T$ as the token that bridges the prompt-continuation boundary, with $s_1$ being the string portion of $T$ in the prompt and $s_2$ being the string portion in the continuation ($\text{decode}(T) = s_1 + s_2$). Because prompt ends at a word boundary by construction, we refer to it as the *word-aligned prompt*. For comparison, we also construct a *token-aligned prompt* by backing off the prompt to the closest token boundary, i.e., $\text{prompt} - s_1$ where we use $-$ to denote removal of a string suffix.

We use two metrics to evaluate model's ability to generate the target continuation from each prompt:

- **Difference in log-probability** that is assigned to the (respective) correct next-token when given the word-aligned versus token-aligned prompt. The correct next-token when conditioning on prompt is $c_1$, i.e., the first element of $\text{encode}(\text{continuation}) = \langle c_1, ..., c_n \rangle$; when conditioning on the token-aligned prompt, the expected next-token is $T$.

$$\Delta\,\text{Logprob} = \log P_M(c_1 \mid \text{encode}(\text{prompt})) - \log P_M(T \mid \text{encode}(\text{prompt} - s_1))$$

- **Difference in accuracy** of greedily decoding the correct continuation when given the word-aligned versus token-aligned prompt. Let $x_1$ be the string continuation given the word-aligned prompt and $x_2$ be that given the token-aligned prompt (in practice, we decode 3 tokens which is empirically sufficient). Then

$$\Delta\,\text{Acc} = \frac{1}{N}\sum_{i=1}^{N} \mathbb{1}[s_2 \sqsubseteq x_1] - \frac{1}{N}\sum_{i=1}^{N} \mathbb{1}[\text{decode}(T) \sqsubseteq x_2]$$

where $N$ is the number of test instances, $\mathbb{1}$ is the indicator function, and $\sqsubseteq$ denotes a string prefix. In other words, the generation $x_1$ given the word-aligned prompt should cover the remainder $s_2$ of the truncated token $T$, and the generation $x_2$ given the token-aligned prompt should cover the entirety of $\text{decode}(T) = s_1 + s_2$.

## 4.3 RESULTS

**Model performance drops substantially under TBP**  Shown in Table 3, all models are severely affected by the TBP, placing less probability on the correct next-token (negative $\Delta\,\text{Logprob}$) and predicting the correct continuation less often (negative $\Delta\,\text{Acc}$) when the prompt boundary does not line up with a token boundary. The difference is striking: on Chinese, for instance, QWEN3-32B places four *orders of magnitude* less probability on the correct next-token and its accuracy drops by 34.98%. Indeed, the accuracy of all models drops by at least 30%.

| Model | $N$ | $\Delta$ Logprob | $\Delta$ Acc (%) |
|---|---|---|---|
| **Chinese** | | | |
| QWEN3-32B | 2304 | $-9.59$ | $-34.98$ |
| HUNYUAN-4B-PRETRAIN | 3551 | $-7.92$ | $-30.75$ |
| **German** | | | |
| QWEN3-32B | 493 | $-7.16$ | $-40.16$ |
| LLAMA-3.1-8B | 513 | $-4.75$ | $-33.92$ |
| MISTRAL-NEMO-2407 | 527 | $-4.01$ | $-48.58$ |
| OLMO2-1124-7B | 491 | $-3.44$ | $-32.38$ |
| GEMMA-3-4B-PT | 534 | $-4.75$ | $-45.13$ |
| **Code** | | | |
| QWEN3-32B | 5223 | $-8.64$ | $-65.82$ |
| LLAMA-3.1-8B | 5863 | $-8.50$ | $-62.00$ |
| MISTRAL-NEMO-2407 | 4729 | $-11.95$ | $-71.05$ |
| OLMO2-1124-7B | 5863 | $-5.88$ | $-58.62$ |
| GEMMA-3-4B-PT | 2325 | $-0.77$ | $-33.94$ |

Table 3: **When facing the tokenization boundary problem, all evaluated models suffer a significant drop in prediction accuracy and probability on the correct continuation.** $\Delta$ **Logprob** measures the average difference in log probability assigned to the correct next-token given the test prompt (*word-aligned*) versus the prompt when backed off to the nearest token boundary (*token-aligned*). $\Delta$ **Acc (%)** measures the average difference in accuracy predicting the correct next-token given the word-aligned vs. token-aligned prompt. We find that when facing the TBP, models place at least two *orders of magnitude* less probability on the true continuation and predict it correctly 30% less often (in absolute terms). $N$ is the size of the test dataset for different tokenziers.

**Scaling does not mitigate the problem**   For model families where multiple model sizes are available, we plot the relationship between model size (in number of parameters) and our two metrics, $\Delta$ Acc and $\Delta$ Logprob. Shown in Figure 2, there is an inverse scaling trend along both metrics for all model families: the performance degradation caused by the tokenization boundary problem is larger in larger models. We hypothesize that this is because larger models are better fit to the tokenizer that was used in training. Given the context $\langle\_\mathtt{process},\mathtt{in}\rangle$, it is more confident that the next token is *not* $\mathtt{g}$ because the sequence $\langle\_\mathtt{process},\mathtt{in},\mathtt{g}\rangle$ was never seen in training (it simply cannot be produced by the tokenizer); the token $\_\mathtt{processing}$ would have observed instead. This suggests that scaling cannot be relied on to resolve this issue; fortunately, we identify a solution in the following section.

## 5  OVERCOMING THE BOUNDARY PROBLEM AT INFERENCE TIME

Finally, we explore whether proposed methods to address the tokenization boundary problem are empirically effective on our data. In particular, we consider the recently proposed `ByteSampler` (Hayase et al., 2025), which solves the boundary problem by constructing a tree of all possible valid token sequences where *(i)* the prompt is a prefix of the decoded sequence, and *(ii)* the last token begins before the prompt ends. Paths in this tree (from root to leaf) correspond to token sequences consistent with the prompt, and are not constrained to having a token boundary at the end of the prompt. We select the path with the greatest cumulative log-probability. After this, we can extend the token sequence by sampling normally, since the sequence ends on a token selected by the model.

### 5.1  SETUP

In §4, we saw that the *word-aligned* prompt (which ends with a complete word but partial token) consistently produces poorer performance than the *token-aligned* counterpart (which backs off to the nearest token boundary). Here, we test whether invoking `ByteSampler` with the word-aligned prompt will recover the desired performance. Like in §4, we consider the prediction accurate if the decoded byte string covers the remainder $s_2$ of the truncated token $T$. We focus on QWEN3-32B, which is the only model we evaluate across all three domains in the main experiments.

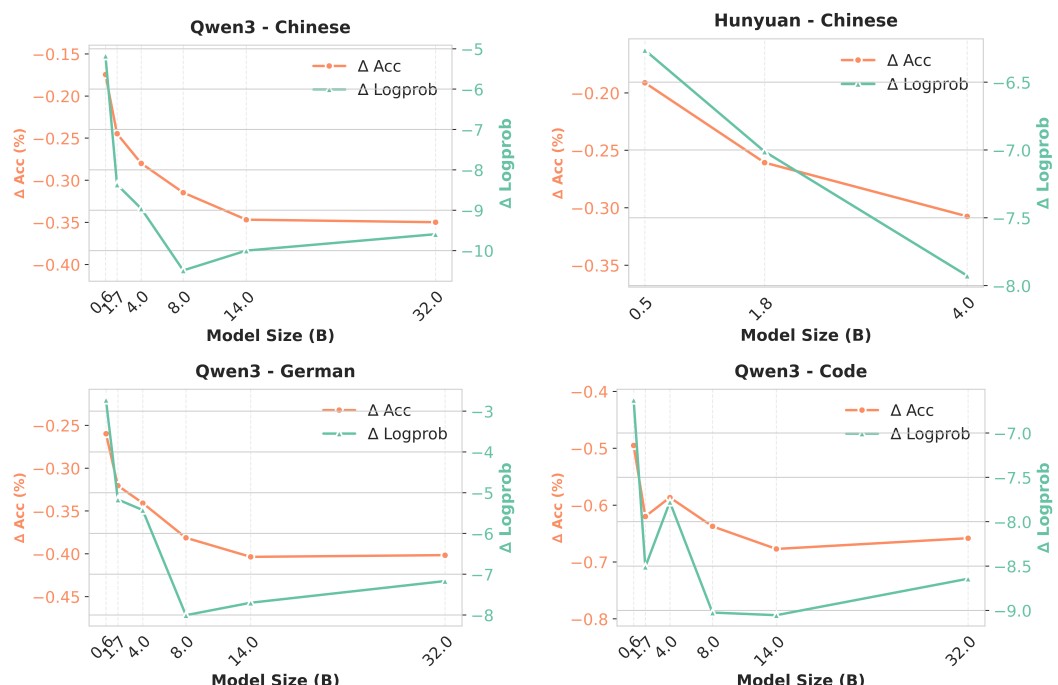

Figure 2: **Sensitivity to the TBP exhibits inverse scaling across different model families**, i.e., larger models are more affected by unexpected token boundaries caused by the prompt ending.

## 5.2 RESULTS

Results are shown in Table 4. Applying `ByteSampler` to the word-aligned prompt not only matches but *exceeds* the performance of the token-aligned prompt. This makes sense, as the word-aligned prompt provides strictly stronger conditioning than the token-aligned prompt, i.e., it is longer by at least one additional word, while asking for the completion of the same overall string. `ByteSampler` is also efficient, requiring on average 0.12 – 1.17 additional forward passes compared to regular sampling.

| Setting | Acc (%) | $\Delta$ | Overhead |
|---|---|---|---|
| **Chinese** | | | |
| Token-aligned | 50.95 | | 0 |
| Word-aligned | 16.18 | −34.98 | 0 |
| Word-aligned + `ByteSampler` | 74.17 | +23.22 | 0.65 |
| **German** | | | |
| Token-aligned | 55.57 | | 0 |
| Word-aligned | 15.41 | −40.16 | 0 |
| Word-aligned + `ByteSampler` | 73.22 | +17.65 | 1.17 |
| **Code** | | | |
| Token-aligned | 71.02 | | 0 |
| Word-aligned | 5.20 | −65.82 | 0 |
| Word-aligned + `ByteSampler` | 86.71 | +15.69 | 0.12 |

Table 4: **Applying `ByteSampler` at inference-time completely solves the tokenization boundary problem, while requiring minimal computational overhead. Acc (%)** is exact-match continuation accuracy; **Overhead** is the average number of additional forward passes used over normal sampling. Results are for QWEN3-32B.

## 6 RELATED WORK AND DISCUSSION

**Tokenization boundary problem**   While the TBP is well-known in the literature, its impact is generally not felt in either English left-to-right generation nor in chat applications (because special tokens separate the prompt and response). Nonetheless, we believe that a better understanding of and solution to the TBP unlocks more future possibilities in tokenization and LM applications. For instance, recent "superword" tokenization methods (Liu et al., 2025; Schmidt et al., 2025), which improve compression by including multi-word tokens, may encounter the TBP even in English use.

Methods proposed to address the TBP (discussed in §2) operate at sampling-time, without requiring changes to model training. Another possible approach is using *stochastic tokenization* in training (Provilkov et al., 2020; Sims et al., 2025; Cognetta et al., 2024) to expose the model to multiple possible segmentations of the same text. These methods are mainly motivated by improving models' subword understanding, however, and its utility for addressing the TBP remains unknown. DEEPSEEK V3 (DeepSeek-AI, 2025) hints at this possibility — it randomly splits some proportion of multi-punctuation tokens into smaller tokens in training explicitly to address the TBP, though they do not present experiments with this ablation. Performing stochastic tokenization more broadly, however, would significantly hurt compression and may bring other side effects, such as diluting the probability of a string over many different tokenizations (Song et al., 2024). We thus recommend pairing the current practice of deterministic tokenization with `ByteSampler` at inference time.

**Token-word misalignment**   It is well-known that tokenizers, especially when learned with the BPE algorithm (Sennrich et al., 2016), generally do not respect derivational, compound, or morphological boundaries within words that are apparent to humans (Chai et al., 2024; Minixhofer et al., 2023). As a result, there have been a substantial number of efforts towards linguistically-informed tokenization (Klein & Tsarfaty, 2020; Hofmann et al., 2021; 2022; Yehezkel & Pinter, 2023; Bauwens & Delobelle, 2024; Si et al., 2023). However, less attention has been paid to the effect this has on generation quality at the prompt boundary.

**Robustness to tokenization**   Our work also adds to the continuously evolving discussion on the extent to which LMs are limited by their tokenization. While it is commonly argued that tokenization obscures orthographic information of tokens (Edman et al., 2024; Chai et al., 2024; Wang et al., 2024), other evidence suggests that LMs naturally learn the characters that make up tokens, especially at scale (Kaushal & Mahowald, 2022; Itzhak & Levy, 2022; Feucht et al., 2024; Kaplan et al., 2025). Recent related work has also shown that models retain semantic understanding of non-canonical tokenizations of the context (Zheng et al., 2025; Geh et al., 2025; Kaplan et al., 2025), for instance, being able to recognize that the character-level tokenization $\langle \text{\textvisiblespace}, \text{c}, \text{a}, \text{t} \rangle$ corresponds to "*\textvisiblespace cat*", even though the token sequence was provably never seen in training (e.g., the existence of the merge $(\text{a}, \text{t})$ implies the tokens $\text{a}, \text{t}$ can never appear in that order). Note that this does not conflict with our findings: one is about understanding non-canonical tokenizations in the context history, whereas our work is about *generating* a non-canonical tokenization.

## 7 CONCLUSION

Language models are designed to provide probability distributions over strings, but are engineered to learn distributions over sequences of tokens. As a result, they model not only language but also the tokenizer that they are trained with. In this work, we revisit a fundamental problem this causes: the ending of a user-provided prompt forces a token boundary that biases the prediction of the model. We demonstrate that this problem appears commonly in natural use cases in languages where whitespace does not reliably delimit semantic units, and that resulting predictions are significantly degraded as a result. Fortunately, we then find that a recently proposed inference-time solution solves the boundary problem completely and efficiently. We hope that our work deepens understanding of the interaction between tokenization and language modeling.

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
