# OpenReview forum: "Are you going to finish that? A Practical Study of the Tokenization Boundary Problem"
_ICLR.cc/2026/Conference — Submitted to ICLR 2026_

### Official Review · Reviewer_n46r · 2025-10-29

**Soundness:** 2
**Presentation:** 3
**Contribution:** 3
**Rating:** 4
**Confidence:** 3

**Summary:**

This paper empirically studies the prevalence of the token boundary problem (TBP) in naturalistic datasets (natural language, code) and proposes the ByteSampler method of Hayase et al. (2025) as a fix. Although this problem is not highly prevalent in English, which uses spaces to separate words, the authors show that naturalistic boundaries (separations between words in Chinese, separations between components of compound words in German, and separations between lexer tokens in code) often occur inside of tokens in the canonical token sequence. They then compare prompts that end at naturalistic boundaries vs. token boundaries and show that LLMs consistently assign higher probability to correct completions following the prompt with the token boundary. They then show that ByteSampler improves the performance of naturalistically-aligned prompts even over that of token-aligned prompts.

**Strengths:**

The paper is generally well-written and easy to follow. The problem is well-motivated; I particularly appreciate Section 3. The experiments use a good variety of LLMs and benchmarks. To my knowledge the prevalence of the TBP has not been systematically studied like this before. The results are very positive (but see my concerns about the evaluation in the Weaknesses section).

**Weaknesses:**

1. My biggest issue with the paper is with the choice of evaluation metrics.
   1. For one, the difference in log probability and difference in accuracy metrics only check that the output matches the first token of the expected output while ignoring the rest of the output. Therefore, it's not clear if ByteSampler actually results in better generation quality beyond the first token of the output. Can you explain why you're only interested in checking whether the output matches a small prefix of the expected output?
   1. For difference in log-probability, wouldn't it make more sense to compare the marginalized probabilities of the *character* continuations? As you mention in the introduction, users only deal with character sequences, not token sequences, so the probabilities of the individual token sequences don't really matter to the user. Using the total probability of character sequences would be a better measure of the quality of the LM output. This is especially important for the word-aligned prompt, because the tokenization whose probability you are currently measuring isn't even canonical, and there may be many other non-canonical tokenizations that have comparable probability; the choice of $c_1$ is, in a sense, arbitrary from the point of view of the LLM.
   1. For difference in accuracy, it would be better to estimate the expected accuracy with respect to the LLM's distribution over character strings rather than the accuracy of the greedy output, as LLMs typically generate responses using ancestral sampling. For each prompt type, you could estimate the expected probability of generating a string that matches the expected output by taking several samples using ancestral sampling and seeing what proportion have a decoded character sequence that matches the expected character sequence. You could also use the beam summing algorithm of Vieira et al. (2025).
1. The prompt completion tasks for natural language are adapted into a machine translation task for the sake of making the completion unambiguous, but translation is full of ambiguity.
1. 406-408: I'm skeptical about this explanation. ByteSampler might select a high-probability canonical token sequence that crosses the prompt boundary, but this begs the question of whether it leads to the correct character sequence with higher probability. If it does, it probably has more to do with the fact that the overall token sequence is always canonical. As mentioned above, the metrics do not even test the accuracy of the generated output beyond the first token.

**Questions:**

1. 190: How exactly do you define "syntactic boundaries" for the programming languages? Would this correspond essentially to lexer tokens? For example, would identifiers, string literals, and operators like `+=` and `->` count as whole words in this case?

---

### Official Review · Reviewer_sM7o · 2025-11-01

**Soundness:** 2
**Presentation:** 3
**Contribution:** 2
**Rating:** 2
**Confidence:** 4

**Summary:**

This paper investigates how mismatches between word-level and token-level boundaries distort LLM behavior. The authors show that, unlike English, many pictographic languages (e.g., Chinese), compounding languages (e.g., German), and code frequently exhibit the tokenization boundary problem (TBP)—where a user’s prompt ends mid-token, causing distorted next-token probabilities. They quantify how often such misalignments occur and find them surprisingly common across these domains. Through controlled experiments, they show that word-aligned prompts (ending mid-token) sharply reduce model confidence and next-token accuracy compared to token-aligned ones, with degradation worsening as model size increases. Finally, they apply ByteSampler, a byte-level inference-time fix that reconstructs the most probable valid tokenization for the prompt, fully restoring—and in some cases improving—accuracy with minimal computational cost.

**Strengths:**

The paper does a good job explaining the underlying intuition behind the tokenization boundary problem (TBP), despite the concept itself being somewhat hard to grasp. The experiments and analyses are convincing and clearly demonstrate that TBP is far more prevalent and impactful in non-English languages such as Chinese and German, as well as in code. The empirical framework is well-designed and methodologically sound, providing sufficient evidence that TBP is a real and practically relevant issue that warrants explicit handling during inference.

**Weaknesses:**

- **W1)**: While the paper’s setup effectively exposes the tokenization boundary problem (TBP), the metrics used to measure “accuracy” and “confidence” may not fully capture its true impact. Because the token-aligned and word-aligned prompts correspond to different conditioning contexts, their next-token distributions are not directly comparable. Much of the observed performance gap could stem from the position of the boundary rather than the boundary error itself—for instance, a token-aligned prompt may naturally end at a more semantically complete chunk (making the next token highly predictable), whereas a word-aligned prompt may stop at the end of one word and query the model for the next, where many continuations are possible. This asymmetry can inflate differences in log-probability and exact-match accuracy. I understand that such back-off comparisons are inherent to the problem setup, but complementing these local metrics with higher-level generation evaluations (e.g., QA accuracy, perplexity, or LLM-as-a-judge coherence scores) would provide a more complete picture of how TBP affects overall generation quality.

- **W2)**: The paper would benefit from a broader comparative evaluation of existing approaches designed to mitigate tokenization errors. While ByteSampler is shown to be highly effective, the study does not experimentally compare it against other methods discussed in the related work—such as token healing, prompt back-off, or stochastic tokenization techniques that modify segmentation during training or inference. Including even a limited benchmark or ablation among these alternatives would clarify whether ByteSampler’s advantages stem from its byte-level formulation or simply from being any form of token-repair mechanism. Such comparisons would make the empirical claims more comprehensive and strengthen the paper’s practical recommendations.

- **W3)**: (Minor) I find the definition of misalignment in the code domain somewhat arbitrary compared to the linguistic settings. In natural languages like Chinese or German, using word segmentation to define boundaries is intuitive and well-motivated. However, for code, the authors treat punctuation boundaries (e.g., after ) or :) as syntactic delimiters, and consider tokens that merge these characters (like ():) as misaligned. It is unclear whether these punctuation points meaningfully correspond to how developers or code models perceive “completion units.” Many real pauses in code editing occur within identifiers, function names, or even whitespace—not necessarily at punctuation marks. As a result, the reported misalignment rates for code may be inflated or reflect a somewhat constructed notion of boundary mismatch, rather than a naturally occurring failure mode.

**Questions:**

- **Q1** : As reported by the authors, the code domain exhibits the highest misalignment rate, yet ByteSampler shows the lowest computational overhead there. Could the authors clarify how this overhead is measured beyond simply counting extra forward passes, and whether there is any relationship between the extent of token misalignment and the actual inference cost?

- **Q2** : Since ByteSampler reconstructs the most probable valid tokenization at the byte level, does it ever alter the semantic continuation or bias generation toward higher-likelihood but less diverse completions? In other words, have the authors observed any trade-off between fixing misalignment and reducing sampling diversity or creativity in open-ended generation?

---

### Official Review · Reviewer_ekvq · 2025-11-04

**Soundness:** 2
**Presentation:** 2
**Contribution:** 2
**Rating:** 2
**Confidence:** 5

**Summary:**

This paper is evaluating the strength of the so called pbp.

**Strengths:**

Realistic problem framing. Instead of contrived prefixes, they target natural word/syntax boundaries in Chinese (no whitespace), German (compounds), and code (punctuation)—the exact places real prompts often end.

 They measure misalignment rates between linguistic/syntactic boundaries and token boundaries—e.g., ~14–25% for Chinese word boundaries, and ≥50% for code punctuation across most tokenizers—so the issue is common, not edge-case.

**Weaknesses:**

The paper provides a nice in depth analysis of the scope and dominance of the pbp in natural languages i am not sure that is enough for a publication. It also seems to advertise for another paper the ByteSampler which is  to my understanding a computationally more elegant version of previous algorithms, tbh that seems like lab mates boosting each other, as i would not consider the paper different in terms of finding a solution to the pbp than other earlier papers.

**Questions:**

Could you clarify you claim of novelty, could you make me underhand why you feature the ByteSampler so prominently?

---

### Official Review · Reviewer_gSEh · 2025-11-04

**Soundness:** 3
**Presentation:** 3
**Contribution:** 3
**Rating:** 6
**Confidence:** 4

**Summary:**

This paper quantifies the tokenization boundary problem in realistic prompts across three domains where word/token boundaries misalign. Chinese (logographic), German (compounding), and code. They find 14-25% misalignment in Chinese, similar rates in German/code, and show that when prompts end at word boundaries but mid-token, models place 2-4 orders of magnitude less probability on correct continuations with 30-70% accuracy drops. The problem exhibits inverse scaling as larger models are more affected.

**Strengths:**

While TBP is known, prior work used arbitrary character prefixes. This is the first systematic study formalizing and quantifying severity in natural prompts respecting word/syntactic boundaries, making it practically relevant.

The controlled setup (word-aligned vs. token-aligned prompts with unambiguous continuations via translation tasks) isolates TBP effects cleanly. Metrics (∆Logprob, ∆Acc) directly measure distortion.

Larger models suffer more from TBP, which is an important finding that is theoretically well-explained and has implications for scaling laws

**Weaknesses:**

The setup favors exact-match metrics but doesn't capture open-ended generation quality degradation.

The inverse scaling hypothesis (larger models fit tokenizer better) lacks empirical validation. No probing experiments, no analysis of which layers/components cause the distortion, no investigation of training dynamics.

Only tested on Qwen3-32B for the solution (Table 4). Does it work equally well across all model families, sizes, and domains? The efficiency claim (0.12-1.17 overhead) may vary significantly by model architecture or tokenizer structure.

No confidence intervals, significance tests, or error bars. Dataset sizes vary by tokenizer (Table 3) but no analysis of whether sample size affects conclusions. Single-run results may not be reliable.
 No comparison to other proposed solutions beyond token backoff.

**Questions:**

Can you empirically test the inverse scaling hypothesis? For example, analyze token prediction confidence on valid vs. invalid sequences, or probe what models learn about tokenizer constraints?

What about the other exact methods cited (Vieira et al., Phan et al., Turaga)? Why specifically recommend ByteSampler over alternatives? Are there any tradeoffs?

 Your setup uses exact-match continuations. How does TBP affect open-ended generation quality (fluency, coherence, task completion) in practice?

Could tokenizers be trained to minimize misalignment? For example, Chinese-specific BPE that respects Jieba boundaries, or code-aware tokenization preserving syntactic units?

---

### Meta-Review · Area_Chair_Frhk · 2026-01-05

**Summary:**

The paper studies how tokenizer boundaries can cause issues in modern LMs. While reviewers appreciated the importance of the problem, several critiques were raised:
- Choice of evaluation metrics.
- Restriction to Qwen3 models.
- Restriction to ByteSampler method.
- Lack of analysis of statistical variation.
- Lack of evidence in favor of inverse scaling hypothesis.
- Limited technical scope.

**Reviewer Concerns:**

As there was no apparent author response, we uphold the above concerns.

**Reviewer Scores:**

As there was no apparent author response, we assume the reviewers' scores would be unchanged.

---

### Decision · Program_Chairs · 2026-01-26

Reject